# A single-blind, randomised control trial on the effectiveness of a structured multi component training module for family caregiver of persons with Parkinson's disease: A study protocol

**Nur Izyan Mohd Amin** [1], **Nor Azlin Mohd Nordin**[1] *, **Aniza Ismail**[2], **Sharmila Gopala Krishna Pillai**[1,3], **Hanif Farhan Mohd Rasdi**[1]

**1** Center for Rehabilitation and Special Needs Studies, Faculty of Health Sciences, Universiti Kebangsaan Malaysia, Kuala Lumpur, Malaysia, **2** Department of Public Health Medicine, Faculty of Medicine, Universiti Kebangsaan Malaysia Medical Centre, Cheras, Kuala Lumpur, Malaysia, **3** Centre for Physiotherapy Studies, Faculty of Health Sciences, Universiti Teknologi MARA, Cawangan Selangor, Kampus Puncak Alam, Puncak Alam, Selangor, Malaysia

* norazlin8@ukm.edu.my

## Abstract

### Introduction

Parkinson disease (PD), a neurodegenerative disorder that progresses over time, is steadily growing in number and prevalence worldwide. PD in Malaysia is expected to increase five-fold by 2040 from the existing estimate of 20,000 patients in 2018. Treatment program of PD in Malaysia is rather unstructured, and there is no known comprehensive PD family care-giver training program available to date. To ensure the quality of a program, it must be tested for feasibility, effectiveness and sustainability. This paper describes the protocol of a study that evaluates the effectiveness of a structured, comprehensive training program of family caregiver to persons with PD in comparison to usual care.

### Materials and methods–Study protocol

A total of 60 pairs of persons with PD of stage II and III, and their primary family caregiver will be recruited and allocated into either an experimental or a control group for 12 weeks of intervention. The experimental group will undergo initial training from multi-disciplinary healthcare providers and will be given a physical module containing weekly tasks that must be practised at home. While the control group will receive a usual care. Both groups will be assessed in terms of physical functions, functional mobility, quality of life (QoL), caregiver burden and knowledge using standardised assessment tools namely Movement Disorder Society-Unified Parkinson's Disease Rating Scale (MDS-UPDRS), Timed Up and Go (TUG) test, Parkinson's Disease Questionnaire (PDQ-39), European Quality of Life five-dimensions (EQ-5D), Malay version of Zarit Burden Interview (MZBI) and Knowledge of Parkinson Disease Questionnaire (KPDQ). In addition, the feasibility and sustainability of the

**Data Availability Statement:** No datasets were generated or analysed during the current study. All relevant data from this study will be made available upon study completion.

**Funding:** Author's name is Prof. Madya Dr. Nor Azlin Mohd Nordin. Grant Number is GUP-2021-059. Funder name is Geran Universiti Penyelidikan (GUP), Universiti Kebangsaan Malaysia. https://research.ukm.my/geran-penyelidikan/ The funders did not and will not have a role in study design, data collection and analysis, decision to publish, or preparation of the manuscript.

**Competing interests:** The authors have declared that no competing interests exist.

**Abbreviations:** PD, Parkinson's disease; PwP, Persons with Parkinson's Disease; QoL, Quality of Life; NMS, Non-motor Symptoms.

interventions will be evaluated, alongside its cost-effectiveness based on the average and incremental cost effectiveness ratio. All data will be analysed using descriptive and inferential statistics, particularly mixed model ANOVA.

## Discussion

There is a significant gap in the literature pertaining family caregiver training programs for people with PD. Documented programs are lacking in term of comprehensiveness of content, application approach and the measurement of training outcomes including the program cost-effectiveness. The feasibility and effectiveness of such training program in a Malaysian setting also requires investigation due to differences in living environment, support system and population's perception. This study will assist to fulfil the existing literature gap and demonstrate the potential benefit of caregiver involvement in mediating the care and therapy for PD in the home setting. Optimum knowledge and skills gained through the training are expected to enhance the confidence and ability of the family caregivers and may possibly reduce their perceived caregiving burden.

## Protocol registration

The protocol of this study is registered in the Australian-New Zealand Clinical Trial Registry (ANZCTR) with a registration number ACTRN12623000336684.

## Introduction

Parkinson disease (PD) is a neurodegenerative disorder that progresses in severity over time. Being the second most common neurodegenerative disease in the world, second only to Alzheimer's Disease, PD is predicted to increase in global incidence and prevalence due to population ageing [1]. In Malaysia, the prevalence of PD is also expected to increase five-fold, from the current estimate of 20,000 to 120,000 cases by the year 2040 [2]. With increasing awareness on the quality-of-life aspect of patients care in this country, PD is now a health concern which receive greater attention.

Because of the disease complexity and its progressive nature, PD commonly cause negative effect on quality of life (QoL) [3]. In a local study in Malaysia, it has been found that PwP experienced significantly lower QoL in all physical and mental health dimensions compared to healthy people [4]. Both motor and non-motor symptoms complicate daily activities of the persons with PD (PwP), alongside the adverse effect of medications, and sedentary lifestyle [4, 5]. Motor symptoms, or physical impairments in PD, which include bradykinesia, rigidity, tremor and postural instability frequently affect mobility; an individuals' ability to move in space and between locations, facilitating their active daily living (ADL) activities across different environments [6]. Mobility limitations among PwP lead to a diminished sense of worth, accompanied by anxiety and frustration [7]. These emotional implications of PD in turn further increase mobility limitation due to low motivation for movement and physical activities.

The symptoms of PD and the negative consequences of physical dysfunctions on daily activities and overall well-being which worsen as the disease progresses commonly result in increasing caregiver burden, and eventually affect their health state or QoL [5]. In the majority of PwP in Malaysia, the primary caregiver is an immediate family member, either the spouse or one of the PwP's children. Their involvement in the caregiving process would exponentially

increase as the PD progresses. As such, evidence-based training program to prepare the caregivers and improve their knowledge and skills in managing PwP are necessary to help reduce the caregiving burden [8, 9].

Past studies have documented positive outcomes of training PD caregivers in their care provision role which include improved knowledge and skills of the caregivers, improved health state of both the caregivers and the PwP and improved functionality of the PwP due to improved quality of care provided by the caregivers [10–13]. Although one may see training family caregivers may further increase the perceived burden experienced by the caregivers due to having to learn, practise and apply various skills on the PwP under their care, a few past studies reported that by providing valuable information and skills, the caregivers feel supported in their day-to-day responsibilities, thus reducing their perceived burden [13]. The increase in confidence level in handling the PwP under care as a result of the training also assists in lowering anxiety and emotional distress due to poor caregiving knowledge. However, currently available studies of PD caregivers training are inadequate in the comprehensiveness of the training content. To be effective, the training programs should involve a systematic and organized series of activities aiming at assisting PwP and caregivers to address issues related to the physical, psychological, and social aspects of their lives [14], and with the goals of managing PD symptoms, enhancing daily functioning, and alleviating burdens of the care and therapy. Such programs should also target to enhance family dynamics which proven to influence how PD caregivers cope and ultimately facilitates in improving the caregivers' understanding, ability to manage situations and overall sense of significance [15]. In addition, the ability of the caregivers to mediate the care or therapy in home setting also worth investigating because reliance on in-hospital therapy by healthcare professionals is not practical in many developing countries with shortage of human resource including Malaysia.

Further, the financial burden faced by patients and caregivers must be given attention to when planning a training program to ensure overall program success and sustainability. The feasibility and effectiveness of a training program should align with its value for money benefits [16]. Review of literature found that, although studies assessing the cost-effectiveness of PD exercise interventions are available [17, 18], the cost-effectiveness evaluation on involving caregiver to mediate PD intervention remains lacking, including in Malaysia.

Therefore, this study is intended to evaluate the effectiveness of a structured multi-component family caregiver training program on physical function and functional mobility of PwP, caregiver knowledge, perceived caregiver burden, and the QoL of both the PwP and the caregiver. We also aim to analyse the cost-effectiveness of the intervention as compared to usual care. We hypothesize that the 12-week structured multi-component family caregiver training program is more effective in improving the targeted outcomes and would be more cost-effective than the usual care.

## Methods

### Study design

This study utilises a single-blind Randomised Controlled Trial (RCT) design. Fig 1 shows a schematic outline concerning enrolment, allocation, intervention, and assessment through the study. The trial will compare two parallel intervention groups, namely the experimental group (structured multi-component family caregiver training program) and the control group (usual care as commonly administered in clinical practice within Malaysian healthcare). The interventions will commence for 12 weeks, with a follow-up at 6 months post-intervention. The flow of the study is shown in Fig 2.

| | STUDY PERIOD | | | | |
|---|---|---|---|---|---|
| | Enrolment & allocation | Baseline | Intervention | Post-intervention assessment | Follow up |
| **TIMEPOINT** | **June 2023 – June 2024** | **0** | **1- 12-weeks** | **week 13** | **24 weeks** |
| **ENROLMENT** | | | | | |
| **Eligibility screen** | X | | | | |
| **Informed consent** | X | | | | |
| **Allocation** | X | | | | |
| **INTERVENTIONS** | | | | | |
| *Family Caregiver Training Program* | | | X | | |
| *Usual Care* | | | X | | |
| **ASSESSMENTS:** | | | | | |
| **Sociodemography** | | | | | |
| Age | | X | | | |
| Gender | | X | | | |
| Education | | X | | | |
| Occupation & income | | X | | | |
| **Primary measure** | | | | | |
| Functionality of PwP (MDS-UPDRS Part III) | | X | | X | X |
| **Secondary measure** | | | | | |
| Functional mobility of PwP (TUG test) | | | | | |
| QoL of PwP (PDQ-39 & EQ5D) | | X | | X | X |
| Caregiver knowledge level (KPDQ) | | X | | X | |
| Caregiver burden (MZBI) | | X | | X | X |
| QoL of Caregiver (EQ5D) | | X | | X | X |
| Feasibility (semi-structured interview, safety and adverse events, adherence, IAM-AIM, FIM) | | | X | X | |
| Cost of intervention (ABC costing) | | | X | X | |

**Fig 1. SPIRIT flow diagram for the schedule of enrolment, interventions, and assessments.** Abbreviations: MDS-UPDRS–The Movement Disorder Society- Unified Parkinson's Disease Rating Scale; Timed Up and Go (TUG) test; QoL–Quality of life; PwP–Persons with Parkinsons; Parkinson's Disease Questionnaire (PDQ-39) EQ5D – European Quality of Life five-dimensions; MZBI–Malay version of Zarith Burden Interview, KPDQ–Knowledge of Parkinson's Disease Questionnaire, IAM-AIM-FIM—the Intervention Appropriateness Measure (IAM), Acceptability of Intervention Measure (AIM), and the Feasibility of Intervention Measure (FIM).

## Study setting and samples

This study will be conducted mostly at the Hospital Canselor Tuanku Muhriz (HCTM), Universiti Kebangsaan Malaysia and the Malaysian Parkinson Disease Association (MPDA) premise for the recruitment, baseline and outcome assessments of the participants. As for the interventions, the location is designed to be the participants' home for the experimental group, while for the control group participants (usual care), the location is not determined by the researchers but would depend on the type of care decided by the patient's treating physician. As an example, if the patient is referred for physiotherapy, the location for intervention would be an out-patient physiotherapy unit of a hospital or a rehabilitation center in the community.

The main sampling frame for this trial is the PwP registered with the MPDA or referred to the PD clinic of HCTM and their respective main family caregiver. The MPDA registry and patient's registry which is maintained by the PD clinic will be accessed to identify potential

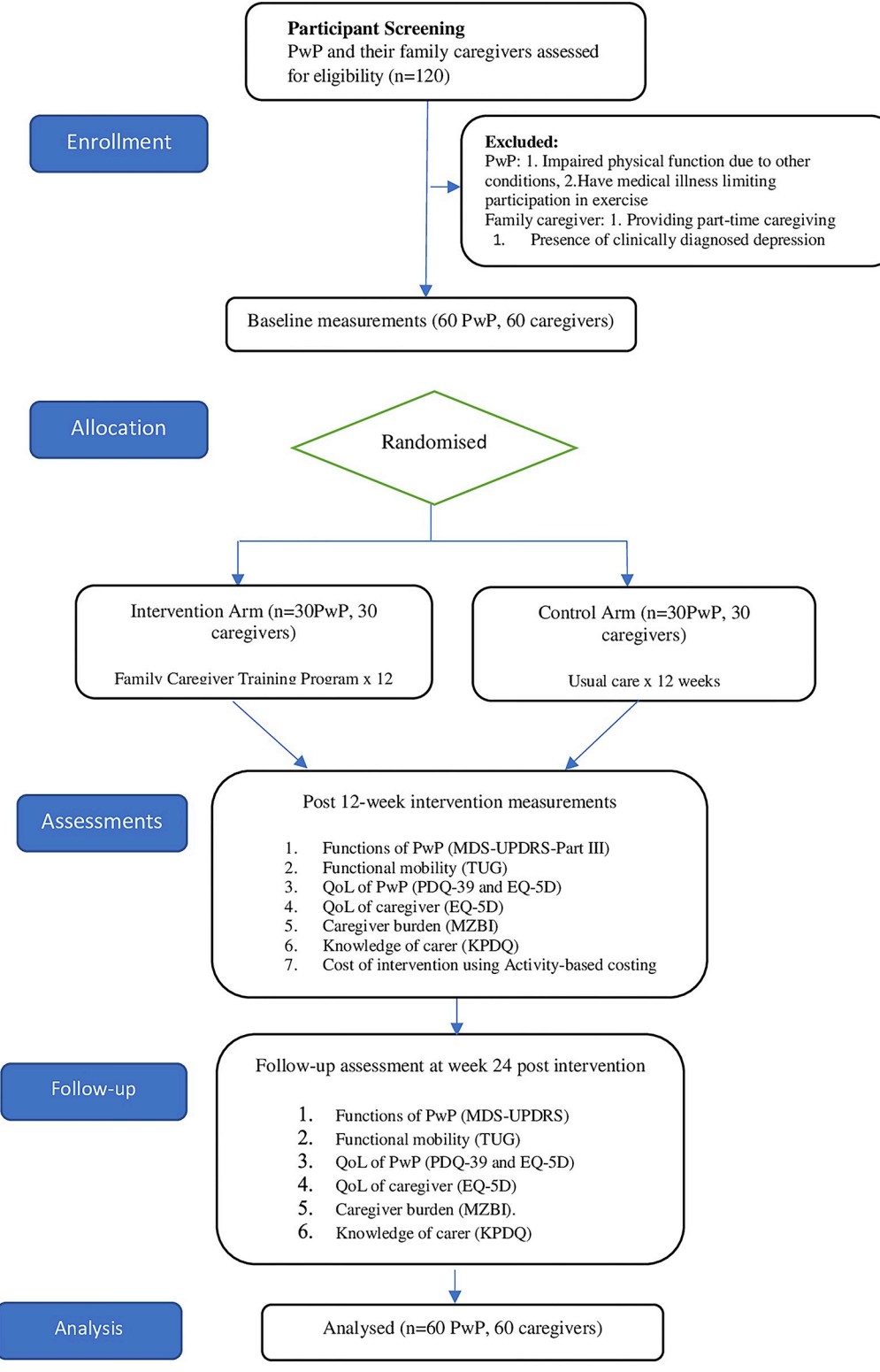

**Fig 2. Consort diagram of the study.**

study participants. Then, a member of the research team will conduct screening of the potential participants using a specified selection criterion. In addition, potential participants will be recruited through advertisements, with the assistance from the MPDA staffs. This is targeting PwP who have not registered with MPDA or not receiving medical care under the PD clinic of HCTM.

**Inclusion and exclusion criteria for PwP participants.** Adults with the following inclusion criteria are eligible to participate; 1) age 30 years or more with a diagnosis of Parkinson Disease stage II-III according to Hoehn & Yahr scale (Table 1), (2) able to provide informed consent, (3) able to walk independently with or without walking aids and 4) must be able to complete questionnaires in English or Malay language. Excluded are PwP with 1) impaired physical function due to other conditions that cause limited walking ability (i.e. muscular dystrophy, a recent fracture within the past 6 months, severe arthritis), or other neurological disease (i.e. peripheral neuropathy, spinal cord injury, stroke); (2) any medical illnesses that may limit participation in exercise (i.e. heart failure, unstable angina, uncontrolled hypertension).

**Inclusion and exclusion criteria for caregivers.** Eligible caregivers of PwP participants must be 1) the main caregiving provider who currently live with and interacts with the PwP daily and 2) able to complete questionnaires in English or Malay language.

The exclusion criteria for the caregivers include 1) providing part-time caregiving due to availability of a maid/helper, 2) presence of clinically diagnosed depression with a cut-off point of $\geq 16$ in the Center for Epidemiologic Studies-Depression (CES-D) and 3) presence of cognitive impairment with a cut-off point of $\leq 26$ in the Montreal Cognitive Assessment (MoCA), as measured by assessor during screening.

**Sample size estimation.** The sample size was calculated using G*power software version 3.1 for ANOVA repeated measures, within-between interactions. This study aims for a power of 80%, with an $\alpha = 0.05$, and an effect size of 0.22 for the primary clinical outcome of MDS-UPDRS part III, subscale motor score. Effect size was derived from the past study by van der Kolk and co-authors (2019) [19] and determined based on the formula for undetermined effect size as documented in Kang (2021) [20], which is using post intervention mean and standard deviation of MDS-UPDRS part III of the home-based and remotely supervised aerobic exercise training for PD (21·2 ±2·0) and control group (20·3 ±2.0). Power calculations indicate total sample size of 36 PwP are needed (i.e. 18 PwP per group). However, to account for the requirement for costing analysis (minimum of 30 subjects per group), a total of 30 PwP per group (60 PwP in total) will be recruited. The 60 PwP will be accompanied by their respective main caregiver, therefore 120 participants (60 pairs of PwP and caregivers) will be recruited. Dropouts are not considered in the sample size calculation because intention-to-treat analysis will be used and all participants who are enrolled at baseline will be included in the final analysis.

## Procedure

Potentially eligible participants will be contacted through text message or phone calls to brief them about the study and obtain consent. A physical meet-up session will be arranged

**Table 1. The Hoehn and Yahr scale for PD staging.**

| Stage | Hoehn and Yahr Scale Description |
|---|---|
| 1 | Unilateral involvement only usually with minimal or no functional disability |
| 2 | Bilateral or midline involvement without impairment of balance |
| 3 | Bilateral disease: mild to moderate disability with impaired postural reflexes; physically independent |
| 4 | Severely disabling disease; still able to walk or stand unassisted |
| 5 | Confinement to bed or wheelchair unless aided |

according to their preferred location either at HCTM or MPDA premise. Participants and their primary family caregiver who agree to participate will be requested to sign a consent form prior to enrolment in the study. Further screening will be conducted using the Hoehn & Yahr scale to confirm the stage of PD of PwP, and CESD and MoCA for the caregivers' emotional and cognitive status, respectively during the meet-up session. Once confirmed as eligible, the PwP and their caregiver will undergo baseline assessments.

### Group allocation & blinding

Participants will be randomised into either the experimental group or the control group using stratified block randomisation after obtaining the informed consent and baseline measurements using 1:1 allocation ratio. Stratifications are based on two age categories (<65 years, ≥65 years), baseline Hoehn and Yahr scale dichotomised into two groups (stage 2, stage 3) and two categories of falls risk using Timed Up and Go (TUG) test score (<11.5s, ≥11.5s). Randomisation will be performed using random allocation software version 1.0, a free open-source web service which generate randomisation sequence. An independent researcher will perform the randomisation and concealed allocation. The implementation of the study involving patient enrolment and administration of intervention will be done by a trained therapist. Another trained therapist, external to the study and blinded to the group allocation will conduct the baseline and post intervention assessment. All participants will be provided with unique identification number to ensure anonymity.

### Interventions

**The experimental group.** Participants randomised into the experimental group will receive a structured, multicomponent and comprehensive family caregiver training program. A recently developed module which is written in Bahasa Malaysia will be used in the training. The training module has six chapters; (i) Introduction to PD, (ii) Guide to medication and general care in PD, (iii) Guide to diet and nutrition (iv) Guide to psychological health (v) Guide to speech and communication, and (vi) Guide to physical health. Table 2 highlights the overview of family caregiver training module to be used in this study.

Training to the participants will be provided by a team of qualified healthcare professionals with experience in their area of expertise. Training will be provided through three training sessions with a maximum of two hours duration for each session. Training of skills will include demonstration by the healthcare professionals and return-demonstrations by the family caregivers. Family caregivers will also be provided recording of the training sessions as reference during the implementation of the 12-week intervention session.

Trained family caregivers will then provide the intervention in the form of education, dietary monitoring, and exercise; physiotherapy, speech therapy and psychology intervention to the PwP guided by the given training module booklet and the recorded training video for 12 weeks. Module booklet consists of clear description of tasks with pictorial guide and detailed instructions for each activity. Weekly phone call monitoring and activity logbook will be used as a form of monitoring of adverse events and participants' adherence.

**The control group.** Participants in the control group will receive the usual care for PwP as commonly practised in Malaysia. The plan of care is normally decided by the treating neurologist. Aside from pharmacological management, PwP may receive non-pharmacological treatments such as one-to-one physiotherapy and/or occupational therapy, and other form of therapy such as speech and psychology intervention if the need arises and depending on the discretion of the treating neurologist. Non-pharmacological management can range from once per week to once a month with commonly one-hour duration each session, and

**Table 2. Overview of the family caregiver training module.**

| Chapter | Subsections | Description |
|---|---|---|
| Chapter 1: Introduction to Parkinson's Disease | 1.1 What is PD<br>1.2 How PD Happens<br>1.3 Main symptoms of PD<br>1.4 Phases of PD<br>1.5 General guide for caregivers of PwP | Participants will be educated by a neurologist on the causes, motor, and non-motor symptoms, the five phases of PD and the general guide for caregivers. |
| Chapter 2: Guide to Medication and General Care in PD | Introduction<br>Health Issues in PD<br>Roles of Healthcare Professionals<br>Information on PD medications<br>2.5 Advanced Therapies | Education on this chapter will be provided by a neurologist. Participants will learn about the health issues in PD and the role of healthcare professionals in providing care for PwP. Benefit and side effects of medications will be provided plus the available advanced therapies like deep brain stimulation for eligible PwP. Participants will be required to fill in the medication diary to track the frequency of medication intake daily for 12 weeks. |
| Chapter 3: Guide to Diet and Nutrition | 3.1 Introduction<br>3.2 Risk Factors of Malnutrition<br>3.3 General Dietary Guidance<br>3.4 Nutrition Guide and Brain Health<br>3.5 Foods to be Reduced in PD<br>3.6 Common Issues and steps to overcome | A dietitian will provide dietary education and the risk factors of malnutrition in PD. Along with general dietary guidance, the dietitian will explain the types of food which optimize brain health plus food to be consumed in limited amounts. Common issues faced by PwP will be addressed and techniques to overcome them from the dietary perspective will be educated. Participants will need to fill in the daily food diary for 12 weeks. |
| Chapter 4: Guide to Psychological Health | 4.1 Introduction<br>4.2 Psychological Health Problems<br>4.3 Psychological Intervention | Psychology education on the common psychological problems faced by PwP, and skills training on the intervention will be provided by a qualified psychologist. Psychological intervention including diaphragmatic breathing technique will be demonstrated while the instructions for mindfulness, imagery and progressive muscle relaxation will be provided in an audio format. Participants will be required to fill in the psychological health intervention log for 12 weeks if psychological intervention are performed. |
| Chapter 5: Guide to Speech and Communication | 5.1 Introduction<br>5.2 Guide to Speech Volume Exercises<br>5.3 Guide to Voice Tone Exercises<br>5.4 Guide to Clear Speech<br>5.5 Guide to Swallowing | Participants will learn from a qualified speech and language pathologist on common speech and communication problems, guide to clear speech, swallowing and controlling saliva. Skills training include exercise demonstration to improve speech volume and voice tone. Participants will be required to fill in the speech exercise logbook for 12 weeks if speech exercises are performed. |
| Chapter 6: Guide to Physical Health | 6.1 Introduction<br>6.2 Physical Health Guide for PwP (Stage I-III)<br>6.3 Physical Health Guide for PwP (Stage IV)<br>6.4 Physical Health Guide for PwP (Stage V)<br>6.5 Guide to Improving Active Daily Living | Two physiotherapists and one occupational therapist will provide education and skills training to the participants in physical health. Exercises description (benefits, techniques, and progression) and pictorial guide will be provided in the module. Recorded videos of the exercises will be provided to participants. Participants will be required to fill in the exercise logbook for 12 weeks. |

organised at either the rehabilitation unit of the hospital or a rehabilitation centre in the community by trained therapists. Similar to the experimental group, participants in the control group will be provided with an activity logbook to document all the care/therapy received and completed, and receive a weekly phone call follow-up to monitor their adherence to the given intervention and any arising adverse effects. Table 3 summarises the interventions to be received by both groups.

**Assessment of treatment feasibility.** Feasibility outcomes will be categorised into feasibility of the training and feasibility of the family caregiver-mediated intervention in this study [21].

*Feasibility of the training.* Feasibility of training is described as the feasibility of the developed module and the training provided by the field expert to family caregiver. This will be

**Table 3. Comparison between experimental group and control group intervention.**

|  | Experimental Group | Control Group |
|---|---|---|
| Characteristics | Family caregiver training program | Usual care |
| Total duration | Average duration of 60–90 minutes per day, three times a week for a total of 12 weeks | Individualized for 12 weeks. Physiotherapy and occupational therapy session (if referred) are usually provided for 60–90 minutes per session, once per week. |
| Caregiver Training provided | Yes | Usually limited to observing/supervising PwP doing home exercise program. |
| Main intervention provider | Family caregiver of PwP | Respective healthcare professionals. |
| Intervention content | Family caregiver training program. Trained family caregivers by field expert in return train the PwP by providing activities comprising education, exercises and various therapy for 12 weeks guided by the developed module. | Individualized treatment provided primarily by a neurologist and for non-pharmacological management, commonly include physiotherapy plus occupational therapy, speech therapy and psychology sessions (if required) as referred by the treating neurologist. |
| Location of intervention | Home setting | In hospitals/rehabilitation centers or community clinic where the PwP obtain medical attention by the treating doctor |

evaluated through semi structured interview with PwP and caregiver. Semi structured interview will consist of questions on (i) understanding and comprehensibility of the module, (ii) experience in filling the logbook, (iii) training delivery method, frequency and intensity by the field-experts, (iv) participants satisfaction towards the developed module and the expert training.

Feasibility of intervention.

- Semi structured interview will be used to evaluate the feasibility of intervention, referring to the 12-week caregiver training program delivered by caregiver to PwP guided by the developed structured multicomponent module. Open-ended questions on suitability, safety and PwP adherence of intervention plus their engagement during the tasks will be recorded.

- Incident reporting which evaluates the safety and adverse event occurred during the intervention will be recorded during control calls and scored 0 for no incidences; 1 for some incidence but solved; 2 for incidences preventing follow ups [22].

- Adherence rate will be measured through the rate of completed sessions. This is achieved by dividing total number of sessions completed over total number of sessions prescribed [21]. Activity logbook documenting all activities completed by PwP will be used for the evaluation, which will then be coded using Likert scale. PwP not performing any activity or exercise will be scored 0, PwP completing some exercises will be scored 1, score 2 will be provided for PwP performing all prescribed exercises and score 3 will be awarded if PwP perform any additional activity exceeding expectations. Completion of 80% of the session indicates high adherence meanwhile PwP performing less than 20% of sessions considered non-adherence [22].

- Acceptability of Intervention Measure (AIM), Intervention Appropriateness Measure (IAM) and Feasibility of Intervention Measure (FIM) will be used to measure the Acceptability, Appropriateness and Feasibility of the intervention [23, 24]. Each measure has four items with a total of 12-items scored on five-point Likert scale, ranging from score 1 = completely disagree to score 5 = completely agree. Higher average scores indicate higher implementation success of the intervention [23].

**Assessment of intervention outcome.** After recruitment, a comprehensive assessment protocol is applied, where baseline measurements will be obtained. PwP are evaluated on their

motor functions, functional mobility, and QoL level. Caregiver will be assessed on their level of caregiver burden, QoL, and knowledge of PD. Participants will undergo this assessment at baseline, post week-12 intervention, and at follow up at 24 weeks (i.e. after six months of the end of the interventions). Assessors will be blinded to intervention and group allocation of each participant during the outcome assessments. Assessment for each pair of participants is split into three sessions for better arrangement. In the first session, assessment of baseline characteristics is taken as sociodemographic data of the sample. This includes information on age, gender, education level, occupation, medical history and relationship between PwP and caregivers. During the second session, assessment of PwP takes place, with involvement of the caregivers. The third session comprises of assessment of caregivers.

**Primary clinical outcome.** *Motor functions*. The primary outcome will be the motor functions of PwP measured using the Part III of MDS-UPDRS, motor subscale. Part III is an investigator rated section which consists of 18 questions of motor examination with five response option, ranging from 0 (normal) to 4 (severe). The higher total summed score indicates greater functional decline among PD persons. Part III of MDS-UPDS has excellent concurrent validity ($r = 0.96$) and high internal consistency ($\alpha = 0.93$) [25].

**Secondary clinical outcomes.** Secondary outcomes are functional mobility of PwP, QoL of PWP and caregiver, caregiver's burden and caregiver knowledge on PD.

*Functional mobility of PwP*. Timed Up and Go (TUG) test will be used to measure functional mobility among PwP. A chair will be placed on one end of the 3 meters (m) and a cone on the other end. PwP will be asked to stand and walk for 3m, turn around the cone, walk again and sit down. Stopwatch will be started as PwP rises from the chair and will be stopped as participants sit down. TUG has excellent test-retest reliability in PwP with intraclass correlation coefficient (ICC) ranging from 0.80 [26] to 0.85 [27]. Scoring of TUG test is the time taken in seconds, (s) to complete one lap as instructed. PwP will be asked to perform TUG test thrice to obtain the average duration.

*QoL of PwP and caregiver*. QoL of PwP will be measured using PDQ-39 and EQ-5D meanwhile QoL of caregivers will be determined using EQ-5D only. PDQ-39 is a disease specific, self-reported questionnaire by PwP on their health related QoL (HRQoL). This questionnaire consists of 39 questions across eight domains namely "mobility", "emotional well-being", "stigma", "social support", "cognition", "communication", and "bodily discomfort" [28]. The response option is a 5-point Likert scale assessing the frequency of impairment ranging from "0 = Never" to "1 = 4 = Always". The minimum score of 0 indicating higher QoL meanwhile a maximum score of 100 represents lower QoL, as higher scores indicating more frequent impairment experienced by PwP [29].

EQ-5D is a standardized, non-disease-specific tool to measure health states comprising of EQ-5D-5L descriptive system and the EQ visual analogue scale (EQ VAS) component (Euro-QoL Research Foundation) [30]. The EQ-5D-5L comprises of five questions of the health domains (mobility, self-care, usual activities, pain/discomfort, and anxiety/depression) with five levels of severity for each question ranging from "no problems" (score = 1) to "extreme problems" (score = 5). The selected digits for all five questions will be combined into a five-digit number describing the participant's health state to obtain health utility index (HUI) based on specific value sets of respective country [30]. The utility score ranging between '0' and '1' represent death and perfect health respectively [31]. Malaysian value set of EQ-5D-5L will be used in this study [32]. EQ VAS, a 20cm scale will record the patient's perceived health status in between two endpoints labelled "worst health" and "best health". The marked value on the scale will then be written in the available box [30].

*Caregiver burden*. The Malay version of Zarit Burden Interview (MZBI) with good psychometric properties will be used to evaluate the caregiver burden [33]. MZBI is a 22-item

caregiver reported questionnaire with 5-point Likert scale option ranging from 0 (Never) to 4 (Nearly always). The maximum total score based on summation of scores of all items is 88 and categorised into four divisions: "score 0–20 = little or no burden, 21–40 = mild to moderate burden, 41–60 = moderate to severe burden, and 61–88 = severe burden" [33].

*Knowledge of caregivers.* Knowledge of Parkinson's Disease Questionnaire (KPDQ) developed by Tan et al (2015) which is also translated in the Malay version will be used to evaluate the knowledge of caregiver on PD in this study. The questionnaire consists of two parts; part one (14 questions) which evaluates the ability of caregiver in recognizing the symptoms of PD; and part two consisting of ten statements with 'true' or 'false' response option to evaluate the general knowledge on PD. Successful identification of PD symptoms and number of questions answered correctly in part two indicates higher level of PD knowledge [34].

**Cost-effectiveness of the intervention.**   Cost-effectiveness of the interventions will be determined by interpreting Average Cost Effectiveness Ratio (ACER) and Incremental Cost Effectiveness Ratio (ICER). In calculating ACER and ICER, the total cost of each intervention will first be estimated, followed by calculating quality adjusted life years (QALY) gained in each intervention group.

*Estimation of intervention cost.* The cost estimation of the interventions employed in this study will utilize mainly the activity-based costing (ABC) method. The costing analysis will be from the societal perspective and comprise two main costs: provider costs and patient costs.

*Provider costs.* Provider costs in this study context are costs to provide the training program or usual care to the PwP and caregiver during the 12-week intervention. These costs encompassed three subcategories: personnel/human resource costs, recurring costs and capital costs. Personnel costs will be computed based on the remuneration of each staff category who are involved in both the interventions, including salaries, allowances, overtime pay, and annual bonuses. These figures are then divided by 12 to determine monthly earnings. Monthly earnings are further divided by 8640 to establish the staff cost per minute (approximating a working schedule of 18 days per month, 8 hours per day, excluding weekends, public holidays, and annual leave). This cost per minute value will then be multiplied with the total duration (in minutes) spent in either training and monitoring participants (for the experimental group) or providing care/therapy (for the control group) during the 12 weeks interventions. For recurring costs such as disposable items, the number of items used in the implementation of intervention in both groups is multiplied by the cost of each item. Also, for recurring utility costs such as telephone bills, it depends on the number of phone calls made by the researchers assigned to monitor the participants. Capital costs, namely the costs of buildings and vehicles used by providers, are not considered in the costing analysis for both groups because the intervention venue could not be standardized for all participants in both groups. However, the cost of rehabilitation equipment potentially used in both groups will be included and the current cost determined after considering the purchase cost and annual depreciation rate of 10%.

*Patient costs.* Patient costs are assessed using ABC method and will be obtained through interview with the PwP and their caregiver. These costs will include both direct costs and indirect costs incurred during the 12-week therapy. Included activities are activities during the training sessions and all activities to be conducted in the PwP home setting through the 12-week intervention following the training. Reference to the participant logbook will be made to identify all activities actually performed at home and the related expenses on the patient/caregiver for each activity. Direct costs comprise transportation cost for visits to either the hospital or MPDA center to attend the training (experimental group) or to rehabilitation center/hospital/clinic (control group), expenses for meals consumed during the visits, fees for registration and services, and other miscellaneous expenditures such as therapy materials or medication/supplement through the 12-week interventions. Whereas, indirect costs will include

**Table 4. Cost components to be included in the intervention cost estimation.**

| Component | Cost Variables | Key items |
|---|---|---|
| **Provider's Costs** | Personnel/Human resource | Staff Salaries |
| | Capital Investments | Therapy Equipments |
| | Recurring Expenses | Telephone bills |
| **Patient's costs** | Transportation | Petrol costs/Public transport fares |
| | | Tolls |
| | | Parking fees |
| | Meals consumed during visits | Food and drinks |
| | Fees for services | Clinic registration costs |
| | | Rehabilitation session costs |
| | | Purchase of rehabilitation/therapy materials |
| | Miscellaneous expenditures | Extra medication/supplements |
| | Loss of income | Patient's / Caregiver's salaries |

loss of income for both the patient and/or their caregivers due to having to attend hospital/ rehabilitation center/clinic visits either for training (in the experimental group) or for therapy (in the control group) and will be estimated based on salary per minute multiplies number of working minutes lost for each visit. Loss of income due to having to mediate care or therapy at home, particularly in the experimental group will not be considered in the patients' costs estimation because the interventions are expected to be mediated outside the working hours. Table 4 summarizes the cost components which will be included in the cost estimation of the intervention in both groups.

The total cost of each intervention will be determined by combining the provider and patient costs. The mean cost per patient for the 12-week intervention can then be calculated and compared.

*Calculation of QALY*. HUI value which will be generated based on EQ5D5L scoring of the participants will be used to calculate QALY for each study participant [35]. The QALY serves as a measure that accounts for both the quality of life (QoL) related to health and the duration of life equally. With QALYs, each year of life post-diagnosis is adjusted based on its quality; one year of perfect health is assigned a value of '1', while a year of less than perfect health is assigned a value of '<1'. Generally, QALYs are calculated by multiplying the HUI score by the remaining number of life years.

*ACER and ICER*. The average cost-effectiveness ratio (ACER) of each intervention is calculated by dividing the mean costs of the intervention by mean QALY gained (QALY at the end of week 12 minus QALY at baseline) among the participants in the group.

$$\bullet\ ACER(experimental) = \frac{Cost(experimental)}{QALY(experimental)}$$

$$\bullet\ ACER(control) = \frac{Cost(control)}{QALY(control)}$$

The incremental cost-effectiveness ratio (ICER) represents the average incremental cost associated with 1 additional unit of the measure of effect. ICER is calculated by dividing the difference in cost between the two intervention groups with the difference in QALY (gained) between the two groups. The Malaysian Gross Domestic Produce (GDP) in Ringgit Malaysia (RM) for the year 2023 will be used in the interpretation of the ICER; any value of 1GDP or

less demonstrates that the intervention is cost-effectiveness.

$$\bullet\ ICER = \frac{Cost(experimental) - Cost(control)}{QALY(experimental) - QALY(control)}$$

**Statistical analysis.** Analysis of interventions outcome will be performed using the 'intention to treat' approach, i.e., all participants who are randomised at baseline will be examined within the assigned group at the end of the intervention. For participants who are unable to complete the intervention and outcome assessments, the 'last observation carried forward' method will be utilised in which their baseline data will be regarded as outcome data. Baseline comparability between the two groups will be assessed prior to finalising statistical tests. SPSS version 23.0 will be used to input and analyse all collected data. Socio-demography data will be analysed descriptively and results presented as mean (standard deviation) or median (IQR) and n(%). Effects of the 12-week interventions and effects at 24-month follow up will be analysed using Mixed Model ANOVA, meanwhile independent sample t test will be used to compare intervention cost and ACER between the two groups. The level of significance is set at p<0.05. Mixed model ANOVA will provide results for time effect, group effect, and time-group interaction effect. Statistically significant time effect and group effect indicates significant within-group changes and between-group difference respectively, while significant time-group interaction effect determine if the interventions yield the desired effect on the dependent variables. The effect size of the interventions on each outcome variable will be analysed by looking at partial eta square ($\eta p^2$) value, and interpreted as large ($\eta p^2 \geq 0.14$), medium ($0.14 > \eta p^2 \geq 0.06$) and small ($\eta p^2 \leq 0.05$) [36]. The results of this study will be reported in accordance with the CONSORT 2010 guidelines.

**Ethical consideration.** The study protocol was approved by the Research Ethics Committee of Universiti Kebangsaan Malaysia (reference number UKM/PPI/111/8/JEP/-2022-302). This trial also received protocol registration approval from the Australian New Zealand Clinical Trials Registry (ACTRN12623000336684). All participants (PwP and their family caregivers) will be given an information sheet, and informed consent will be obtained prior to enrolment into the study.

## Discussion

In this article, we describe the protocol of a single blind, randomized control trial to determine the clinical and cost-effectiveness of a structured, multi-component family caregiver training program in comparison to a usual care for PwP population.

Review of literature found several PD family training programs implemented in past studies which focused on caregiver skills training [10, 11] or educational sessions [12]. In general, these programs showed promising results in assisting PwP and their family caregiver to achieve better self-management. However, the available training program presented in the past studies is lacking in comprehensiveness, did not include input from multidisciplinary healthcare professionals which is important to address the complex care needs of the PwP and did not evaluate caregivers' ability in completing the care in home settings for a specified time. In addition, no study has been attempted to develop and test PD family training program in Malaysia to date, although family plays an important role in supporting PwP in the Malaysian culture.

Furthermore, caregiver burden is a topic often discussed in the management of a long-term diseases, like PD, due to the changing demands as the condition progresses. Numerous researches have been done to investigate the effect of PD caregiving on the physical, emotional

and social wellbeing of the caregivers as a whole [11, 37–39]. In a past study, it has been stated that a lack of knowledge and not knowing what to expect following the disease diagnosis is a source of burden to the caregivers as there are many uncertainties that come with the disease [12]. By having knowledge, skills and expectations on how to overcome each of the challenges that may be faced, caregiver will be more prepared and have a better strategy to handle situations and tasks. Family training programs have shown to give positive results in terms of physical and emotional status. This is evident in a 2019 study that utilised an occupation-based Parkinson caregiver program [13], which has noted improvements on specific components of the QOL scale.

Our proposed study aims to demonstrate the outcomes among PwP and their family caregivers who receive a comprehensive training combining both the specified skills and education by a team of multidisciplinary health professionals in managing the complex manifestations in PD. The results of this study are expected to contribute to the pool of evidence regarding PD training programs which utilizes home-based intervention and caregiver involvement. In the long-term it is hoped that this type of intervention will help to consolidate a sustainable care network, which will fulfil the multiple needs of people with PD and their families.

## Supporting information

**S1 Checklist. SPIRIT 2013 checklist: Recommended items to address in a clinical trial protocol and related documents*.**
(DOC)

**S1 File.**
(DOCX)

## Acknowledgments

We thank Universiti Kebangsaan Malaysia for the ethical approval and funding of this study (Study code GUP-2021-059).

## Author Contributions

**Conceptualization:** Nur Izyan Mohd Amin, Nor Azlin Mohd Nordin.

**Data curation:** Nur Izyan Mohd Amin, Sharmila Gopala Krishna Pillai.

**Formal analysis:** Sharmila Gopala Krishna Pillai, Hanif Farhan Mohd Rasdi.

**Funding acquisition:** Nor Azlin Mohd Nordin.

**Investigation:** Nur Izyan Mohd Amin, Sharmila Gopala Krishna Pillai.

**Methodology:** Nur Izyan Mohd Amin, Nor Azlin Mohd Nordin, Aniza Ismail.

**Supervision:** Nor Azlin Mohd Nordin, Aniza Ismail.

**Validation:** Nor Azlin Mohd Nordin.

**Visualization:** Nor Azlin Mohd Nordin.

**Writing – original draft:** Nur Izyan Mohd Amin.

**Writing – review & editing:** Nor Azlin Mohd Nordin, Aniza Ismail.

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
