## [Decision Letter · Decision Letter 0]

30 Jan 2024

PONE-D-23-30057A Single-Blind, Randomized, Controlled Trial Evaluating the Effectiveness of a Structured, Multi-Component Training Module for Family Caregivers of Persons with Parkinson’s diseasePLOS ONE

Dear Dr. Mohd Amin,

Thank you for submitting your manuscript to PLOS ONE. As you will see below, whilst recognising the considerable  value of a study on this topic, the reviewers had considerable concerns about the protocol - as do I. Therefore, after careful consideration, whilst we feel that the protocol has merit, it does not fully meet PLOS ONE’s publication criteria as it currently stands. Unfortunately, I realise it may now be too late to make all the changes that would adequately address these concerns, as you have already started the trial. However, should you believe that you are able to fully address all the points made during the review process, then you are welcome to submit a revised version of the manuscript for further consideration.

Please submit any revised manuscript by Mar 15 2024 11:59PM. If you will need more time than this to complete your revisions, please reply to this message or contact the journal office at plosone@plos.org. Please include the following items when submitting your revised manuscript:A rebuttal letter that responds to each point raised by the academic editor and reviewer(s). You should upload this letter as a separate file labeled 'Response to Reviewers'.A marked-up copy of your manuscript that highlights changes made to the original version. You should upload this as a separate file labeled 'Revised Manuscript with Track Changes'.An unmarked version of your revised paper without tracked changes. You should upload this as a separate file labeled 'Manuscript'.

We look forward to receiving your revised manuscript if you feel this is appropriate.

Kind regards,

Antony Bayer

Academic Editor

PLOS ONE

Journal Requirements:

2. Please include “Protocol” in the manuscript title.

5. Please amend the manuscript submission data (via Edit Submission) to include author Dr. Hanif Farhan Mohd Rasdi.

Additional Editor Comments:

In general the protocol needs to be much more precise and detailed. Using the protocol, it should be possible for a reader to be able to conduct the trial identically at another site. Terms such as "such as" (16 times) are not appropriate.

More information is needed to explain and justify the sample size. It seems rather optimistic by comparison with other similar trials. Was the patient population in ref 15 (Bueno) the same as proposed in this trial? Ref 16 included patients with HY stage 1-3.  How was the "effect size of 0.41" derived? If a significant clinical benefit can be shown with 36 dyads, is it appropriate to recruit another 24 merely to have 60 dyads to enable economic benefits to be assessed (or is a 40% drop out expected?) No detail is given about the cost analysis or how the relevant data for this will be collected.

There should be only one primary outcome. If TUG is chosen, then why would any difference not be due to general fitness secondary to the regular caregiver training rather than something specific to PD?  If UPDRS is the primary outcome, how does this impact on sample size? Why was an outcome such as PDQ-39 or NMSQ not considered? 

Why only HY stage 1-2? Such patients are only minimally impaired & unlikely to show large deterioration within 6 months, meaning it is likely to be very challenging to show benefit of intervention over control. Is follow up long enough?

Other considerations include: Is the intervention instead of, or in addition to usual care? How will blinding of the outcome assessments be assured? Will patients' medication be optimised before randomisation? Why is there an upper age of 75? Is this clinically (and ethically) justified?

The numbering of the references seems to have gone wrong and the reference citation is not consistent. The names of authors also need to be checked (e.g. Zarit not Zarith)

Reviewers' comments:

Reviewer's Responses to Questions

**Comments to the Author**

1. Does the manuscript provide a valid rationale for the proposed study, with clearly identified and justified research questions?

Reviewer #1: Partly

Reviewer #2: Partly

2. Is the protocol technically sound and planned in a manner that will lead to a meaningful outcome and allow testing the stated hypotheses?

Reviewer #1: Partly

Reviewer #2: Partly

3. Is the methodology feasible and described in sufficient detail to allow the work to be replicable?

Reviewer #1: No

Reviewer #2: No

4. Have the authors described where all data underlying the findings will be made available when the study is complete?

Reviewer #1: No

Reviewer #2: No

5. Is the manuscript presented in an intelligible fashion and written in standard English?

Reviewer #1: Yes

Reviewer #2: Yes

6. Review Comments to the Author

You may also provide optional suggestions and comments to authors that they might find helpful in planning their study.

Reviewer #1: This is a very important and significant study.

Some suggestions for the authors;

1. Under Introduction section, have an "Objectives section" to make the primary and secondary objective distinct.

2. More details re randomisation method, i.e. who generated the randomisation schedule? Allocation ratio, ?variable block randomisation etc...

3. In the Data analysis, this is an RCT mention that results will be reported in accordance with the CONSORT 2010 guidelines

4. Define intention to treat population.

5.Finalisation of analysis plan prior to unblinding of the data.

6. Briefly Mention handling of missing data?

7. What about non-compliance or adherence, consideration of how this will be handled

8. Any consideration of sensitivity analysis, i.e to be detailed in analysis plan, but perhaps to consider impact of non-adherence on treatment effect.

9. Mentioning drop-outs due to intention to treat may not be appropriate. Intention to treat relates to including all in the analysis regardless of the randomised group and regardless of non-compliance. Consideration of drop-outs (line 226) in studies is to make sure that the study is powered enough to ensure that you can still answer the research question, please consider revising this statement,

Reviewer #2: This study aimed to design a structured, multi-component training program for caregivers to implement at home for patients with PD. While the aim of the study may have its merit, there are multiple study design details need to be clarified in order to determine the novelty as well as importance of the study.

Introduction

1. In the introduction, it is not clear why the authors want to focus the structured training programs for caregivers. Some discussion should be added regarding the potential advantages and disadvantages of having the caregivers to train people with PD.

Methods

1. Page 6, line 157-170: the study hypothesis should be moved to the Introduction section, following research aim.

i. If the primary goal is to reduce cost, then cost-effectiveness analysis should be placed as the primary outcome.

2. Page 8-9:

i. Why focus on patients with PD at Hoehn and Yahr stage I & II? The early stages of patients are still active and can engage in various outdoor activities. Additionally, patients with PD at this stage do not need full-time caregivers to take care of them. Thus, the exclusion criteria of the caregiver in Table 1 seems to be awkward. Why not include patients with PD at Stage III? Patients at later stages would suffer a lot of motor and non-motor symptoms, and would thus significantly impact their ADLs and quality of life. They might also rely more on their caregivers as well as having longer time to stay at home.

ii. Since caregivers will be the primary person providing the intervention for this study, the relationship of the patients with their caregivers should be considered. Additionally, the compliance and education level of the caregivers should also be considered.

3. Page 9-10: The calculation of sample size needs to be revised. Based on my understanding of the study goal and study design, each patient with PD should be matched with 1 caregiver, so the sample size should consider as pairs instead of calculating patients and caregivers as each individual leading to a double size sample. Hence, according to the sample size calculation, 30 pairs of patient-caregiver should be recruited for each group leading to 60 pairs (60 patients and 60 caregivers) in total.

4. Page 12: Explain more detail about the caregiver training sessions, including the number of sessions, the duration of each session, and the way to make sure the caregivers can implement the techniques properly.

5. Page 13: What will be the treatment dosage of the control group? Will it be the same as the Experimental group? In Table 2, there seemed to be not equivalent in the treatment dosage. This will be a huge confounding factor.

6. Table 2: the content that the family members going to apply is too vague. How to make sure whether the family members know what to provide to the patient at what timing?

7. Page 14 outcome measures:

i. Why choose TUG as the primary outcome? The hypothesis put cost-effectiveness as the primary benefit of the experimental group.

ii. The authors talked a lot about non-motor symptoms in the Introduction. However, there are very limited outcomes evaluating non-motor symptoms, except for some parts of UPDRS. I would suggest the authors add more outcomes testing non-motor symptoms for this study.

Discussion

1. Why establishing a protocol in Malaysia is important? Why cannot the authors use the same protocol established in other countries? Are there any specific culture-specific health care issues in Malaysia that require the conduction of this trial?

7. PLOS authors have the option to publish the peer review history of their article (what does this mean?). If published, this will include your full peer review and any attached files.

Reviewer #1: No

Reviewer #2: No

---

## [Author Response · Author response to Decision Letter 0]

30 Jul 2024

Dear PLOS ONE Editorial Office,

We are very appreciative of the valuable feedback provided by the two reviewers. We are grateful for their feedback, which has been a crucial contribution in enhancing this manuscript. We have diligently responded to all their comments which can be found in the following pages. This rebuttal follows the format in which the reviewers’ comments are written according to the original order, in blue, and our responses come directly after each of those points. We have carefully considered the comments and made the necessary revisions to improve the clarity and precision of our protocol. 

The following are responses to comments made by Reviewer #1:

1. Reviewer Comment: In general, the protocol needs to be much more precise and detailed. Using the protocol, it should be possible for a reader to be able to conduct the trial identically at another site. Terms such as "such as" (16 times) are not appropriate.

Author Response: In response to this point, we have added detailed descriptions to all relevant sections of the methodology. Specifically, we have revised the text to provide clearer guidelines and specifications, making it possible for the protocol to be replicated at another site without ambiguity (Page 6, line 145 to page 24, line 581).

Additionally, we have removed vague phrases, such as "such as," and replaced them with specific details where necessary. This change was implemented across the manuscript, including in the SPIRIT flowchart, to ensure consistency and clarity.

2. Reviewer Comments: 

i) More information is needed to explain and justify the sample size. It seems rather optimistic compared with other similar trials. Was the patient population in ref 15 (Bueno) the same as proposed in this trial? Ref 16 included patients with HY stage 1-3. How was the "effect size of 0.41" derived?

ii) If a significant clinical benefit can be shown with 36 dyads, is it appropriate to recruit another 24 merely to have 60 dyads to enable economic benefits to be assessed (or is a 40% dropout expected?)

iii) No detail is given about the cost analysis or how the relevant data for this will be collected.

Author response: We have revised the sample size calculation in our manuscript. The primary outcome measure has been updated to the MDS-UPDRS as suggested by the reviewers. The effect size of 0.41 was derived from a study by van der Kolk et al. (2019), using the method documented in Kang (2021). Specifically, we used the post-intervention mean and standard deviation of the MDS-UPDRS Part III scores for the home-based and remotely supervised aerobic exercise training group (21.2 ± 2.0) and the control group (20.3 ± 2.0). The calculations indicate that a total of 36 pairs of participants are required, equating to 18 pairs in each group, resulting in a total of 72 participants (Page 9, line 217 to page 10, line 238).

For the cost-effectiveness evaluation, a minimum of 30 samples for each outcome is recommended. Considering both patient-related and caregiver-related outcomes, we propose recruiting 30 pairs (30 patients and 30 caregivers) for each group. This results in a total of 60 pairs or 120 participants. This sample size accounts for potential dropout and ensures sufficient power for the economic analysis (Page 9, line 230 to page 10, line 238).

We have now included a detailed methodology for the cost and cost-effectiveness assessment in the manuscript. The cost variables have been itemized and presented in Table 4. We will collect data on direct medical costs, non-medical costs, and indirect costs associated with the intervention and control conditions. This comprehensive approach will enable a robust analysis of the economic impact of the intervention (Page 20, line 458 to page 22, line 527).

3. Reviewer Comment: There should be only one primary outcome. If TUG is chosen, then why would any difference not be due to general fitness secondary to the regular caregiver training rather than something specific to PD? If UPDRS is the primary outcome, how does this impact the sample size? Why was an outcome such as PDQ-39 or NMSQ not considered?

Author Response: We appreciate the reviewer's insights and acknowledge that the Timed Up and Go (TUG) test may not be the most appropriate primary outcome measure for this study. As a result, we have changed the primary outcome to the MDS-UPDRS Part III. The sample size calculation has been adjusted accordingly to reflect this change, ensuring that the study is adequately powered to detect differences in this measure (Page 17, line 385 to line 392).

In addition, we have decided to include the EQ-5D-5L and PDQ-39 as secondary outcome measures. The EQ-5D-5L is necessary for obtaining the health utility index required for the calculation of Quality-Adjusted Life Years (QALYs) in the cost-effectiveness analysis, using the Incremental Cost-Effectiveness Ratio (ICER) formula. The inclusion of PDQ-39 allows for a comprehensive assessment of the participants' quality of life, addressing both motor and non-motor symptoms of Parkinson's Disease (Page 18, line 407 to Page 19, line 436).

4. Reviewer Comment: Why only HY stage 1-2? Such patients are only minimally impaired and unlikely to show large deterioration within 6 months, meaning it is likely to be very challenging to show the benefit of intervention over control. Is follow-up long enough?

Author Response: We appreciate the reviewer's insightful observation. Our initial focus was on caregivers of patients with early-stage Parkinson's Disease (PD), classified as Hoehn and Yahr (HY) stages 1-2. However, upon reviewing our training module and considering the reviewer's comments, we have decided that our intervention is better suited for patients in HY stages 2-3. This adjustment will enable us to target a population with a more pronounced level of impairment, increasing the likelihood of observing significant outcomes within the study period (Page 7, line 191).

Regarding the follow-up duration, we believe that the current follow-up period is sufficient to observe meaningful changes in the primary and secondary outcomes. However, we 

acknowledge the importance of longer-term follow-up and have considered planning future studies with extended follow-up periods to capture more long-term effects of the intervention.

The following are responses to comments made by Reviewer #1:

1. Reviewer Comment: Under the Introduction section, have an "Objectives section" to make the primary and secondary objectives distinct.

Author Response: We have added a paragraph within the Introduction to clearly delineate the primary and secondary objectives of the study (page 5, line 133-141). This section now explicitly outlines our goals and hypotheses:

• Main Objective: To evaluate the effectiveness of the program on physical function in patients with Parkinson's Disease.

• Secondary Objectives: To assess the program's effectiveness on functional mobility, knowledge about the disease, caregiver burden, quality of life (QOL), feasibility of the intervention, cost-effectiveness, and sustainability of the benefits.

• Hypothesis: The program will be more effective compared to usual care in improving the aforementioned outcomes.

2. Reviewer Comment: More details on the randomization method are needed, including who generated the randomization schedule, allocation ratio, and whether variable block randomization was used.

Author Response: We have added a detailed description of the randomization method in the manuscript. The randomization was conducted by an independent researcher using specialized software to ensure unbiased assignment of participants. The allocation was set at a 1:1 ratio for the experimental and control groups. We employed variable block randomization to maintain balance across groups throughout the study. Additionally, we included stratification based on [specify stratification variables if applicable, e.g., disease severity, age, gender] to ensure that important baseline characteristics were evenly distributed between the groups (Page 10, line 255 to page 11, line 270).

3. Reviewer Comment: In the Data Analysis section, this is an RCT. Mention that results will be reported in accordance with the CONSORT 2010 guidelines. Finalization of the analysis plan prior to unblinding of the data. Briefly mention handling of missing data.

Author Response: We have revised the Data Analysis section to include the following details :

• CONSORT 2010 Guidelines: Results will be reported in accordance with the CONSORT 2010 guidelines to ensure comprehensive and transparent reporting of the randomized controlled trial (Page 23, line 557 to page 24, line 582).

• Analysis Plan Finalization: The analysis plan will be finalized prior to unblinding of the data to prevent any potential bias and ensure that the statistical methods applied are pre-specified and rigorous.

• Handling of Missing Data: We have included an explanation of our approach to handling missing data. We will employ an intention-to-treat (ITT) analysis to account for any missing outcome data, ensuring that all randomized participants are included in the analysis as originally allocated. Additionally, we will use appropriate methods for handling missing data, such as multiple imputation or other statistical techniques, to address any gaps and maintain the robustness of our findings (Page 23, line 560).

4. Reviewer Comment: Define intention to treat population.

Author Response: We apologize for the error in terminology. The correct term is "intention-to-treat (ITT) analysis," not "intention-to-treat population." We have corrected this in the manuscript. 

5. Reviewer Comment: What about non-compliance or adherence? Consideration of how this will be handled?

Author Response: Participants with low adherence will be actively followed up, and efforts will be made to provide motivation and support to improve compliance with the intervention. We will implement strategies to address non-compliance, such as regular check-ins, and reinforcement of the importance of adherence to the intervention protocol. These measures are designed to maximize participant engagement and adherence throughout the study (Page 23, line 560).

6. Reviewer Comment: Any consideration of sensitivity analysis, i.e., to be detailed in the analysis plan, but perhaps to consider the impact of non-adherence on treatment effect.

Author Response: Thank you for the suggestion regarding sensitivity analysis. While we do not plan to include a formal sensitivity analysis in the initial protocol, we recognize the value of this approach. Should we observe a substantial rate of non-adherence in our study, we will consider conducting a sensitivity analysis to evaluate the impact of non-adherence on the treatment effect. This analysis will be detailed in any subsequent reports if necessary.

7. Reviewer Comment: Mentioning drop-outs due to intention to treat may not be appropriate. Intention to treat relates to including all in the analysis regardless of the randomized group and regardless of non-compliance. Consideration of drop-outs (line 226) in studies is to make sure that the study is powered enough to ensure that you can still answer the research question; please consider revising this statement.

Author Response: Thank you for pointing this out. We have removed the statement on drop-outs as our study will utilize intention-to-treat (ITT) analysis, which includes all randomized participants regardless of compliance or drop-outs. This approach ensures that all outcome data will be available for analysis and maintains the integrity of randomization. We will ensure that our study is powered adequately to answer the research question effectively (Page 9, line 235).

The following are responses to comments made by Reviewer #2:

1. Reviewer Comment: This study aimed to design a structured, multi-component training program for caregivers to implement at home for patients with PD. While the aim of the study may have its merit, there are multiple study design details that need to be clarified in order to determine the novelty as well as importance of the study.

Author Response: Thank you for your feedback. We have added detailed information regarding the study methodology to clarify the design and implementation of the structured, multi-component training program for caregivers (Page 6, line 145 to line 165). 

2. Reviewer Comment: In the introduction, it is not clear why the authors want to focus the structured training programs for caregivers. Some discussion should be added regarding the potential advantages and disadvantages of having the caregivers train people with Parkinson's Disease.

Author Response: Thank you for your insightful comment. We have added a discussion in the Introduction about the importance of a comprehensive, structured training program for caregivers (Page 4, line 94 to page 5, line 123). This addition includes:

o Empowering caregivers with structured training can improve their ability to provide effective care, enhance the quality of life for both the caregiver and the patient, and potentially lead to better management of Parkinson's Disease symptoms.

o Structured programs can also provide caregivers with valuable skills and resources to support patient care, improve adherence to treatment protocols, and reduce caregiver burden.

3. Reviewer Comment: Page 6, lines 157-170: The study hypothesis should be moved to the Introduction section, following the research aim. If the primary goal is to reduce cost, then the cost-effectiveness analysis should be placed as the primary outcome.

Author Response:

We have moved the study hypothesis to the Introduction section to align with the reviewer’s suggestion and improve the clarity of the manuscript (Page 5, line 133-141).

4. Reviewer Comment: Why focus on patients with PD at Hoehn and Yahr stage I & II? The early stages of patients are still active and can engage in various outdoor activities. Additionally, patients with PD at this stage do not need full-time caregivers to take care of them. Thus, the exclusion criteria of the caregiver in Table 1 seems awkward. Why not include patients with PD at Stage III? Patients at later stages would suffer from a lot of motor and non-motor symptoms, and would thus significantly impact their ADLs and quality of life. They might also rely more on their caregivers as well as having longer time to stay at home.

Author Response: Thank you for your valuable insight. We agree with your suggestion and have revised the study population criteria to include patients at Hoehn and Yahr stages II and III. We have also updated Table 1 to reflect these changes and believe this modification enhances the relevance and applicability of our study (Page7, line 192).

5. Reviewer Comment: Since caregivers will be the primary person providing the intervention for this study, the relationship of the patients with their caregivers should be considered. Additionally, the compliance and education level of the caregivers should also be considered.

Author Response: Thank you for highlighting these important factors. We will collect information on the relationship between patients and their caregivers, including details about the nature and quality of their relationship. Additionally, we will assess caregiver compliance and gather data on their education level and occupation (including income level). This information will help us better understand the dynamics influencing the intervention and its effectiveness (Page 17, line 378 to line 382).

6. Reviewer Comment: The calculation of sample size needs to be revised. Based on my understanding of the study goal and study design, each patient with PD should be matched with 1 caregiver, so the sample size should consider pairs instead of calculating patients and caregivers as individual units, leading to a double-sized sample. Hence, according to the sample size calculation, 30 pairs of patient-caregiver should be recruited for each group, leading to 60 pairs (60 patients and 60 caregivers) in total.

Author Response: Thank you for pointing this out. We have revised the sample size calculation accordingly. The updated calculation now reflects the require

---

## [Decision Letter · Decision Letter 1]

16 Aug 2024

A Single-Blind, Randomised Control Trial on the Effectiveness of a Structured Multi Component Training Module for Family Caregiver of Persons with Parkinson’s Disease: A Study Protocol

PONE-D-23-30057R1

Dear Dr. Mohd Amin,

Thank you for your revised manuscript that now addresses previous concerns (and that will require your trial registration with ANZCTR to be updated). 

We’re pleased to inform you that your manuscript has been judged scientifically suitable for publication and will be formally accepted for publication once it meets all outstanding technical requirements. 

Kind regards,

Antony Bayer

Academic Editor

PLOS ONE

Additional Editor Comments (optional):

Reviewers' comments:

Reviewer's Responses to Questions

**Comments to the Author**

1. Does the manuscript provide a valid rationale for the proposed study, with clearly identified and justified research questions?

Reviewer #1: Yes

Reviewer #2: Yes

2. Is the protocol technically sound and planned in a manner that will lead to a meaningful outcome and allow testing the stated hypotheses?

Reviewer #1: Yes

Reviewer #2: Yes

3. Is the methodology feasible and described in sufficient detail to allow the work to be replicable?

Reviewer #1: Yes

Reviewer #2: Yes

4. Have the authors described where all data underlying the findings will be made available when the study is complete?

Reviewer #1: Yes

Reviewer #2: Yes

5. Is the manuscript presented in an intelligible fashion and written in standard English?

Reviewer #1: Yes

Reviewer #2: Yes

6. Review Comments to the Author

You may also provide optional suggestions and comments to authors that they might find helpful in planning their study.

Reviewer #1: No further comments - all have been addressed.

Reviewer #2: The revised manuscript is much improved from the previous version, and the authors have addressed my previous concerns.

7. PLOS authors have the option to publish the peer review history of their article (what does this mean?). If published, this will include your full peer review and any attached files.

Reviewer #1: No

Reviewer #2: No

---

## [Editor Report · Acceptance letter]

9 Sep 2024

PONE-D-23-30057R1 

PLOS ONE

Dear Dr. Mohd Amin, 

I'm pleased to inform you that your manuscript has been deemed suitable for publication in PLOS ONE. Congratulations! Your manuscript is now being handed over to our production team.

Kind regards, 

on behalf of

Professor Antony Bayer 

Academic Editor

PLOS ONE